# A Subtle Aspect of Minimal Lengths in the Generalized Uncertainty Principle

Michael Bishop [1,†], Joey Contreras [2,†] and Douglas Singleton [2,*,†]

1   Mathematics Department, California State University Fresno, Fresno, CA 93740, USA;
    mibishop@mail.fresnostate.edu
2   Physics Department, California State University Fresno, Fresno, CA 93740, USA;
    mkfetch@mail.fresnostate.edu
*   Correspondence: dougs@mail.fresnostate.edu; Tel.: +1-559-278-5281
†   These authors contributed equally to this work.

**Abstract:** In this work, we point out an overlooked and subtle feature of the generalized uncertainty principle (GUP) approach to quantizing gravity: namely that different pairs of modified operators with the same modified commutator, $[\hat{X}, \hat{P}] = i\hbar(1 + \beta p^2)$, may have different physical consequences such as having no minimal length at all. These differences depend on how the position and/or momentum operators are modified rather than only on the resulting modified commutator. This provides guidance when constructing GUP models since it distinguishes those GUPs that have a minimal length scale, as suggested by some broad arguments about quantum gravity, versus GUPs without a minimal length scale.

**Keywords:** generalized uncertainty principle; quantum gravity; minimal length





## 1. Basic Motivation and Structure for the Generalized Uncertainty Principle

In this opening section, we review the ideas and motivations underlying the generalized uncertainty principle (GUP) approach to quantum gravity. GUP is a phenomenological approach to quantum gravity which introduces an absolute minimal length in the theory. Many different approaches to combining quantum mechanics and gravity are thought to require a minimal length [1–9]. There is a simple physical argument for this. From the Heisenberg uncertainty relationship (i.e., $\Delta x \Delta p \geq \frac{\hbar}{2}$), one sees that quantum mechanics gives the following relationship between uncertainty in position and momentum $\Delta x \sim \frac{Const.}{\Delta p}$. From the gravity side, one argues that as one tries to probe smaller distances, one needs to go to higher center of mass energies/momenta. At some point, the energy/momentum will be large enough that one will form a micro-black hole whose event horizon size can be estimated by the Schwarzschild radius $r_{Sch} = 2G\Delta E/c^2$, whereas in this expression, $\Delta E$ has replaced the conventional mass of the black hole, $M$. Now, further setting $c = 1$, replacing $r_{Sch}$ with $\Delta x$ and $\Delta E$ with $\Delta p$, this relationship becomes $\Delta x = 2G\Delta p$, i.e., there is now a linear relationship between $\Delta x$ and $\Delta p$. It should be noted that since we have a set $c = 1$ mass, the energy and momentum are interchangeable. If one combines this linear relationship from gravity with the inverse relationship from quantum mechanics, one finds $\Delta x \sim \frac{\hbar}{\Delta p} + G\Delta p$, (ignoring the factors of 2). The interplay between the linear term and the inverse term lead to a minimum in $\Delta x$ at $\Delta p_m \sim \sqrt{\hbar/G}$ of $\Delta x_m \sim \sqrt{\hbar G}$.

The minimal $\Delta x$ that comes from the GUP, as described above, is one way to avoid the point singularities that occur in certain solutions in general relativity such as black hole spacetimes. If one cannot resolve distances smaller than $\Delta x_m$, this may lead to the avoidance of the singularities of general relativity. This is the hope for theories of quantum gravity—that they will allow one to avoid the singularities of classical general relativity. Another approach to avoid these singularities of classical black hole solutions is non-commutative

geometry [10]. In this approach, one proposes that coordinates do not commute with one another. For example, $[X, Y] \neq 0$ or $[Y, Z] \neq 0$. We will later show that the types of GUP models favored by our analysis are also connected to non-commutative geometry theories.

One of the strengths of the phenomenological GUP approach to quantum gravity is that it offers the possibility to make experimental tests of quantum gravity—to experimentally check whether there is a minimal distance resolution, $\Delta x_m$, as implied by the above arguments, and if so, what is the size of this minimal distance resolution? Some tests of the GUP scenario rely on astrophysical phenomena. For example, reference [11] proposed a test of minimal lengths based on the dispersion of high-energy photons coming from short gamma ray bursts. The idea of [11] was that having a minimal distance scale in ones' theory would alter the standard energy–momentum relationship of special relativity, $E^2 = p^2 c^2 + m^2 c^4$. This altered energy–momentum relationship would then lead to an energy-dependent speed of light in the vacuum, which in turn would lead to photons of different energies dispersing or spreading out as they travel long distances through the vacuum. In 2009 [12], the Fermi gamma ray satellite detected high energy photons coming from a distant gamma ray burst. Using the analysis of [11], the observation by the Fermi satellite was able to place bounds on the deviations from $E^2 = p^2 c^2 + m^2 c^4$ due to a minimal distance scale. Surprisingly, if the deviations from the special relativistic photon energy and momentum relationship were linear in energy (i.e., $p^2 c^2 = E^2 [1 + \zeta (E/E_{QG}) + \ldots]$ with $E_{QG}$ being the quantum gravity scale and $\zeta$ is a parameter of order 1) then the observations of [12] implied the bound $E_{QG} > E_{Planck}$, i.e., that there was no deviation to energies beyond the Planck energy scale. Or putting this in terms of length $l_{QG} < l_{Planck}$, which is counter to the expectation that hints of quantum gravity should occur before reaching the Planck-scale.

There are also tabletop and small-scale laboratory test proposals for testing for effects connected with the minimal distance scale coming from GUP. In the works [13,14], the proposal was made to use the Lamb shift, Landau levels, quantum tunneling in scanning tunneling microscopes. This work showed than using these tabletop experiments, one could put a bound on the parameter $\beta$ which were, in units of Planck momentum squared, of $\beta < 10^{36}$, $\beta < 10^{50}$ and $\beta < 10^{21}$ for the Lamb shift, Landau levels and tunneling, respectively. There is also a proposal [15] to test Planck scale physics with a tabletop cryogenic optical step-up where the optical photon's momentum is coupled to the center of mass motion of a macroscopic transparent block in such a way that the block is displaced in space by approximately a Planck length. In the works [16–18], it was shown that one could use detailed studies of wavefunctions of large molecule to probe Planck-scale physics. Finally there is recent work [19] which looked at using gravitational waves to put bounds on the parameters coming from GUP models. All of these various experimental approaches are welcome since they hold out the hope that one may experimentally probe Planck-scale physics.

We now briefly review some of the basic background behind GUP models and particularly focus on the role that modified operators have in determining whether or not there is a minimal length. The uncertainty relationship between two physical quantities is closely tied to the commutation relationship between the operators which represent this quantities. In general, for two operators $\hat{A}$ and $\hat{B}$, one has the following relationship between the uncertainties and the commutator:

$$\Delta A \Delta B \geq \frac{1}{2i} \langle [\hat{A}, \hat{B}] \rangle . \tag{1}$$

where the uncertainties are defined as $\Delta A = \sqrt{\langle \hat{A}^2 \rangle - \langle \hat{A} \rangle^2}$ and similarly for $\langle \hat{B} \rangle$. For standard position and momentum operators in position space $\hat{x} = x$ and $\hat{p} = -i\hbar \partial_x$, one obtains the usual commutator $[\hat{x}, \hat{p}] = i\hbar$ which then implies the standard uncertainty principle,

$$\Delta x \Delta p \geq \frac{\hbar}{2}$$

or the operators are $\hat{x} = i\hbar\partial_p$ and $\hat{p} = p$ in momentum space.

To obtain a GUP characterized by $\Delta x \sim \frac{\hbar}{\Delta p} + \beta\Delta p$, [7] proposed the following modified commutator:

$$[\hat{X}, \hat{p}] = i\hbar(1 + \beta\hat{p}^2) . \tag{2}$$

Note that following [7], we replace $G$ by a phenomenological parameter $\beta$ which characterizes the scale where quantum gravity is important. Naively, the quantum gravity scale would be set by $G, \hbar$ and $c$, but in large extra dimension models [20,21] or brane world models [22–26], the quantum gravity scale can be lower than the typical Planck scale. In these brane world models, $\beta$ would be set by the value of the higher dimensional Newton's constant and the size of the extra dimensions. Furthermore, in (2), the position operator is capitalized while the momentum operators, on both the left and right sides of (2), are not. This is because in reference [7], the choice was made that in order to obtain the modified commutator in (2), they would modify the position and momentum operators as

$$\hat{X} = i\hbar(1 + \beta p^2)\partial_p \quad \text{and} \quad \hat{p} = p \tag{3}$$

where only the position operator is modified. This choice of operators in (3) is absolutely crucial to obtaining a minimal length. Using (2) in (1) gives:

$$\Delta X\Delta p \geq \frac{\hbar}{2}(1 + \beta\Delta p^2) \tag{4}$$

In arriving at (4), we are taking the center of mass coordinates with $\langle\hat{p}\rangle = 0$. Dividing both sides by $\Delta p$ gives:

$$\Delta X \geq \frac{\hbar}{2}\left(\frac{1}{\Delta p} + \beta\Delta p\right), \tag{5}$$

so that one arrives at the type of GUP described in the opening paragraph which definitely leads to a minimal length. Minimizing $\Delta X$ in (5) shows it has a minimal length at $\Delta p = \sqrt{1/\beta}$ which then yields $\Delta X_{min} = \hbar\sqrt{\beta}$.

In contrast to the work on GUP from reference [7], as outlined above, most recent works have followed a different approach, such as that found in [27] where the *same* modified commutator of (2) was obtained by modifying the momentum operator *but* not the position operator. By taking the operators to be of the form:

$$\hat{x} = i\hbar\partial_p \quad \text{and} \quad \hat{P} = p\left(1 + \frac{\beta}{3}p^2\right), \tag{6}$$

one can see that plugging these into the commutator immediately leads to the same right hand side of the commutator as in (2) where capitals indicate the operator is modified. However, the uncertainty between the two operators in (6) is subtly different from the two operators in (3). Using the operators in (6) to calculate the uncertainty relationship via (1) gives:

$$\Delta x\Delta P \geq \frac{\hbar}{2}(1 + \beta\Delta p^2). \tag{7}$$

Although this relationship superficially appears to be the same as before, now the left hand side has $\Delta P$ for the modified momentum, while the right hand side has $\Delta p$ for the standard momentum. Thus, instead of (5) one obtains:

$$\Delta x \geq \frac{\hbar}{2}\left(\frac{1}{\Delta P} + \frac{\beta\Delta p^2}{\Delta P}\right). \tag{8}$$

In contrast to (5), it is not clear whether the interplay between $\Delta p$ and $\Delta P$ on the right hand side of (8) will lead to a minimum. If the second term $\beta\frac{\Delta p^2}{\Delta P}$ is increasing with $\Delta p$, then there is a minimum; but if this term decreases with $\Delta p$, then there is no minimum. Since the highest power of $p$ in $\hat{P} = p(1 + \frac{1}{3}\beta p^2)$ is $p^3$, one can show that $\Delta P \approx \beta\Delta p^3$, which

implies that the second term $\beta\frac{\Delta p^2}{\Delta P}$ will go like $\frac{1}{\Delta p}$, i.e., decreasing with $\Delta p$ and thus there is no minimum in position.

To explicitly see the difference between the GUP from (5) versus the GUP from (8), one can plot the uncertainty in position versus the uncertainty in momentum in a family of test functions for each operator pair. In Figure 1, we plot $\Delta X$ versus $\Delta p$ from (5) for $\beta = 0.01$. One can see that there is a minimum in $\Delta X$ around $\Delta p = 1/\sqrt{\beta}$ as expected.

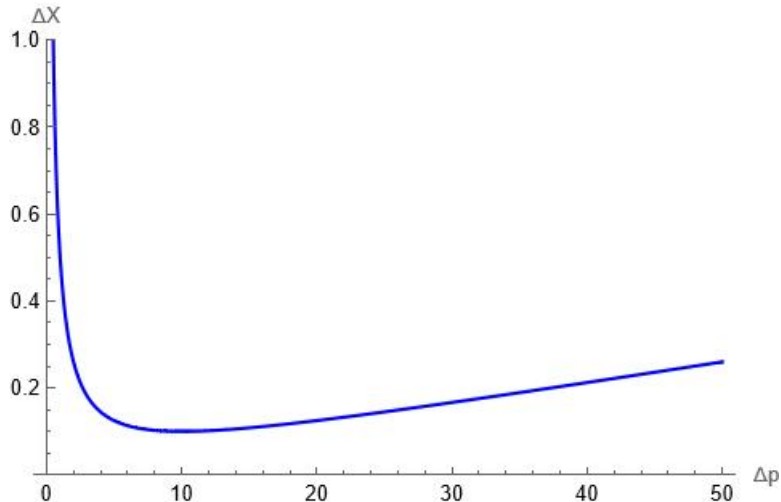

**Figure 1.** The relationship between $\Delta X$ and $\Delta p$ for the GUP from (5) using $\beta = 0.01$. As expected, a minimum length occurs at approximately $\Delta p = 1\sqrt{\beta}$, and after this minimum, $\Delta X$ increases with $\Delta p$.

In Figure 2, we plot $\Delta x$ versus $\Delta p$ from (8) again with $\beta = 0.01$. In contrast to Figure 1, Figure 2 has no minimum $\Delta x$ and in fact looks like the standard relationship between $\Delta x$ and $\Delta p$ from quantum mechanics.

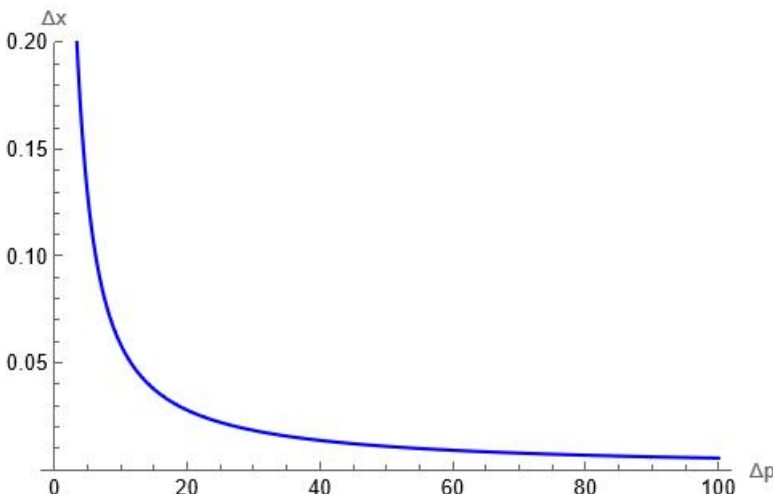

**Figure 2.** The relationship between $\Delta x$ and $\Delta p$ for the GUP from (8) using $\beta = 0.01$. This GUP has no minimal length and essentially behaves in a manner resembling the standard uncertainty principle.

There are some subtleties in how Figures 1 and 2 were obtained. First, for both figures, we used the lower bound (i.e., the equal sign) of Equations (5) and (8). Second, in calculating $\Delta P$ for use with (8), we used a test Gaussian wavefunction in momentum space, $\Psi(p) \propto e^{p^2/2\sigma}$. The reason for using a specific test wave function, rather than making a general argument that would apply regardless of the form of the wavefunction, was to complicate the relationship between $\hat{P}$ and $\hat{p}$. For a GUP like that in (4), one has the same $\Delta p$ on both the left and right hand side of the equation, regardless of the wavefunction.

This is the result of not modifying the momentum operator in this GUP model. On the other hand for a GUP such as (7), the momentum uncertainty is different between the left and right hand sides of the equation, and there is not a simple relationship between $\Delta P$ and $\Delta p$. Thus, for this case, we picked a specific wavefunction to calculate $\Delta P$ and $\Delta p$ and check how the uncertainty principle worked out. One could choose a different test wavefunction instead of the Gaussian, but the results would be quantitatively similar to that shown in Figure 2. Note that the effects for the type of GUP shown in Figure 2 are most pronounced in the $\Delta p \to 0$ limit. The above calculations are a prelude and check for the examples of GUPs given in the following section.

Finally, even though the momentum operator associated with (5) is the standard one from quantum mechanics, the change in the position operator forces one to change the measure of integration in $p$ in order to ensure that $\hat{X}$ and $\hat{p}$ are symmetric [7]. This change in the measure of the momentum integration amounts to $\int \ldots dp \to \int \ldots \frac{dp}{1+\beta p^2}$. This modifies the normalization constants but does not qualitatively affect the behavior $\Delta X$ in the limit as $\Delta p \to \infty$. All of these issues are more fully examined in [28].

The main point of this section is to show that many works on GUP following the seminal work of KMM [7] and make a different choice for how the operators are modified, which does not necessarily result in a minimum length scale. KMM [7] modified the position operator, but left the momentum operator unchanged, whereas many other works chose to modify the momentum but leave the position operator unchanged. Thus, certain choices of modifying the operators are wrong in the sense that they do not lead to a minimal length. From the above, we conclude that, when it comes to determining whether a theory has a minimum length scale, **how the operators are modified is more important than how the commutator is modified**. In short, a modified commutator is insufficient to guarantee a minimum length scale; in fact, a modified commutator is not necessary at all, as shown in the next section.

## 2. A GUP with an Unmodified Commutator

We now move on to give details of how a modified commutator is not necessary to achieve a minimum length scale. If we do not modify the commutator nor modify the operators, we would obviously end up with ordinary quantum mechanics, which does not have a minimal length. Thus, we want modified position and momentum operators $\hat{X}$ and $\hat{P}$ which give rise to the usual commutator $[\hat{X}, \hat{P}] = i\hbar$, [29,30]. This in turn leads to a standard looking uncertainty relationship but now in terms of the modified operators $\Delta X \Delta P \geq \frac{\hbar}{2}$. In order to have a minimum $\Delta X$, one needs to modify $\hat{P}$ so that $\Delta P$ is either capped at some constant value or decreases. In reference [29], a GUP of this kind was constructed by defining a modified momentum by

$$\hat{P} = p_M \tanh\left(\frac{\hat{p}}{p_M}\right), \tag{9}$$

where $p_M$ is the maximum cap on the modified momentum. This maximum momentum, $p_M$, is an upper limit to the momentum in the relationship $E^2 = p^2 + m^2$. This cap in momentum also implies a cap in the energy $E$. This connection between a cap on momentum and a cap on the energy of a single particle, as well as how this relates to modified position and time operators, is discussed in greater detail in [29]. Picking the modified position to have the form:

$$\hat{X} = i\hbar \cosh^2\left(\frac{\hat{p}}{p_M}\right)\partial_p \tag{10}$$

then leads to the standard looking commutator $[\hat{X}, \hat{P}] = i\hbar$ as can be verified by the substitution of (9) and (10) into the commutator. Another variant with a capped momentum is given in [29] by

$$\hat{P} = \frac{2p_M}{\pi} \arctan\left(\frac{\pi p}{2p_M}\right),$$ (11)

and for the modified position operator, one has:

$$\hat{X} = i\hbar\left[1 + \left(\frac{\pi p}{2p_M}\right)^2\right]\partial_p.$$ (12)

Both forms of the modified momenta given in (9) and (11) lead to the result $\Delta P \leq p_M$ which then gives $\Delta X \geq \frac{\hbar}{2p_M}$. Despite modified operators (9) and (10) or (11) and (12) leading to a minimum $\Delta X$, there is a difference with how this is achieved compared to the KMM modified operators of (3). Plotting $\Delta X$ versus $\Delta p$ coming from (9), (10), the associated uncertainty principle gives the result shown in Figure 3; the graph of $\Delta X$ versus $\Delta p$ for the operators (11) and (12) looks very similar. In Figure 1, which shows the plot of the Kempf–Mangano–Mann GUP [7], the minimum in $\Delta X$ is reached at some $\Delta p$ and thereafter $\Delta X$ linearly increases with $\Delta p$. In contrast, the GUP shown in Figure 3 asymptotically approaches the minimum $\Delta X$.

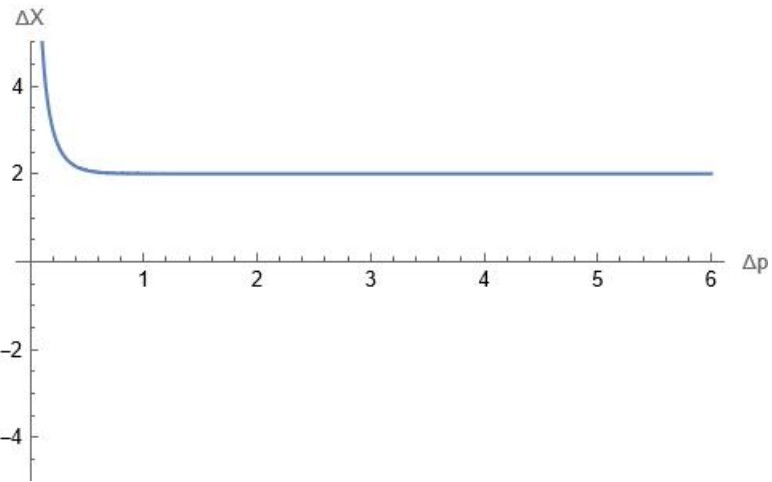

**Figure 3.** The relationship between $\Delta X$ and $\Delta p$ for the modified operators (9) and (10) and the associated GUP using $p_M = 0.25$

The GUP model given by Figure 1 has a physical motivation for its form: $\Delta x$ inversely proportional to $\Delta p$ for small momentum where quantum mechanics dominates and has $\Delta x$ proportional to $\Delta p$ for a large momentum where quantum gravity is thought to dominate. The motivation for this behavior was laid out in the introduction and relied on the formation of micro black holes at large momentum/small distances. In contrast the GUP model given by Figure 3, which has $\Delta x$ inversely proportional to $\Delta p$ for small momentum, but at a large momentum has $\Delta x$ approach a constant value asymptotically, does not have a simple, physical motivation.

The main take-away message of this section was to emphasize that the most important factor in whether or not a given GUP leads to a minimum length is how the operators are modified.

## 3. Connection to Non-Commutative Geometry and Running of $G$

In this section, we tie the above considerations about GUPs to another approach to quantizing gravity and a potential physical consequence of quantum gravity. The other approach to quantizing gravity is non-commutative geometry and the physical consequence is the running of the coupling constant of a theory—in this case, the running of Newton's $G$.

In general, for non-commutative spacetimes, one has a non-trivial commutation relationship between coordinates of the form $[X_i, X_j] = i\theta_{ij}$ where $\theta_{ij}$ is an anti-symmetric matrix [10]. This non-commutativity between coordinates implies an uncertainty relationship $\Delta X_i \Delta X_j \geq \frac{1}{2}|\theta_{ij}|$, which in turn implies that there is a minimal area and volume—one cannot make $\Delta X \Delta Y$ or $\Delta X \Delta Y \Delta Z$ (for example) arbitrarily small due to the non-commutativity between the coordinates. This has the effect, similar to GUP models, of preventing the formation of point singularities that occur in a black hole and other solutions of classical general relativity. It has previously been noted [7] that certain GUP models lead to modified position operators in three dimensions (3D) which naturally result in the non-commutativity of the modified position operators. In one spatial dimension (1D), one does not encounter this non-commutativity since a single operator will always commute with itself. However, in three dimensions, different coordinates may not commute with one another, e.g., $[\hat{X}, \hat{Z}] \neq 0$. This is the reason behind the matrix $\theta_{ij}$ being antisymmetric.

As a specific example of the GUP-non-commutative geometry connection, one can look at the 3D version of the modified position operators of [7]. Letting the position and momentum in (3) go from 1D to 3D (i.e., $\hat{X} \to \hat{X}_i$, $\hat{p} \to \hat{p}_i$ and $\partial_p \to \partial_{p_i}$ gives a 3D version of (3) of the form:

$$\hat{X}_i = i\hbar(1 + \beta|\vec{p}|^2)\partial_{p_i} . \tag{13}$$

With this 3D version of (3), the coordinate commutator becomes:

$$\begin{aligned}
[\hat{X}_i, \hat{X}_j] &= -2\beta\hbar^2(1 + \beta|\vec{p}|^2\partial_{p_j})(p_i\partial_{p_j} - p_j\partial_{p_i}) \\
&= 2i\hbar\beta(\hat{p}_i\hat{X}_j - \hat{p}_j\hat{X}_i) ,
\end{aligned} \tag{14}$$

which implies a connection to the non-commutative parameter of $\theta_{ij} = 2\hbar\beta(\hat{p}_i\hat{X}_j - \hat{p}_j\hat{X}_i)$. As required, this $\theta_{ij}$ is antisymmetric. In the second line in (14), we used (13) to turn this back into an expression in terms of the (modified) position operator and (unmodified) momentum operator. The result in (14) was previously derived in [7].

One can also create 3D versions of the modified position operators from (10) and (12) which take the form:

$$\hat{X}_i = i\hbar \cosh^2\left(\frac{|\vec{p}|}{p_M}\right)\partial_{p_i} , \tag{15}$$

and:

$$\hat{X}_i = i\hbar\left[1 + \left(\frac{\pi|\vec{p}|}{2p_M}\right)^2\right]\partial_{p_i} . \tag{16}$$

Just as in the case of the 3D modified position operators in (13), the modified 3D position operators in (15) and (16) also lead to a non-commutativity of the coordinates, $[\hat{X}_i, \hat{X}_j] \neq 0$. We do not give the specific form of $\theta_{ij}$ for the modified 3D position operators from (15) and (16) since we only want to make the point here that some GUPs (particularly those studied in this work) lead to non-commutative geometry.

In quantum field theory, when interactions are quantized, one encounters the phenomenon that the coupling "constants" of the interaction become dependent on the energy/momentum scale at which the effects of the interaction are measure. Colloquially one talks about coupling constant "running" with the energy/momentum scale. For example, in quantum electrodynamics, the fine structure constant, $\alpha = \frac{e^2}{\hbar c}$, which measures the strength of the electromagnetic interaction, depends on the energy/momentum scale, $E/p$, at which it is measured. Similarly, the weak and strong nuclear interactions have couplings which scale with energy/momentum. One can view the GUP approach to quantum gravity to be as running from Newton's constant $G$. The commutator from (2) implies that the GUP parameter $\beta$ depends on the momentum of the interaction as $\beta(p) = \beta p^2$, i.e., the parameter $\beta$ scales quadratically with momentum in this case.

To translate this running of the phenomenological parameter $\beta$ into a running of Newton's constant $G$. We simply recall that from the heuristic arguments $\Delta x_m \sim \sqrt{\hbar G}$ as well as in terms of $\beta$, one has $\Delta x_m \sim \hbar\sqrt{\beta}$. Thus, we have the connection between $G$

and $\beta$ of $\beta \sim \frac{G}{\hbar}$. Thus, a $\beta$ which "runs" with the momentum directly implies a running $G$. There are two things to note: (i) the usual running of a coupling like the fine structure parameter $\alpha = \frac{e^2}{\hbar c}$ is usually logarithmic in perturbative quantum fields theory; (ii) there are differences between gravity and the other interactions that make it unclear that one can consistently define a running gravitational coupling, at least in the usual approach of quantum fields theory, as detailed in the arguments of reference [31].

## 4. Summary and Conclusions

In this short note, we examined the role that modifying the position and momentum operators plays in determining a minimum length and that focusing on modifying the commutator is insufficient. We examined models that had modified commutators with the same right hand sides, as in (2), but where the operators on the left hand side were different. We considered the case where the position was modified and the momentum remained unchanged (see (3)) and we considered the case where the position remained the same and the momentum was modified (see (6)). The former case led to a minimum length while the latter case did not. This can be explicitly seen by comparing Figures 1 and 2. Finally, we presented a case where the position and momentum were modified, but the commutator remained the same as the standard one from quantum mechanics—Equations (9) and (10). This system show that there can be a minimum length scale without modifying the commutation relation.

A general conclusion that can be distilled from the work in this paper is the following. **In order for a GUP to have a minimal length, the key factor is the modification of the operators rather than the modification of the commutator.**

The thrust of this paper was to argue that there are constraints on the specifics of how one can formulate a GUP to obtain a minimal length. This is welcome since while the phenomenological approach of GUPs has strengths (e.g., the possibility to confront ideas about quantum gravity with experiments and observations), one would also like a way to constrain and focus on the range of variations in the operators and commutator. Other recent works [32,33] have also looked at ways to constrain the form of GUPs.

Finally, in the last section, we tied the type of GUP models discussed here with other models of quantum gravity and with other potential consequences of quantum gravity, namely non-commutative geometry and the "running" of the coupling strengths of interactions.

**Author Contributions:** M.B., J.C. and D.S. each equally contributed to all aspects of this research work. All authors have read and agreed to the published version of the manuscript.

**Funding:** This research received no external funding.

**Institutional Review Board Statement:** Not applicable.

**Informed Consent Statement:** Not applicable.

**Data Availability Statement:** Not applicable.

**Acknowledgments:** All the authors acknowledge discussion with Jaeyeong Lee in early parts of this work.

**Conflicts of Interest:** The authors declare no conflict of interest.

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
