# Peer review of "A Subtle Aspect of Minimal Lengths in the Generalized Uncertainty Principle"

_universe, doi:10.3390/universe8030192_

Round 1

Reviewer 1 Report

The authors present an overview of a subtle feature of the Generalised Uncertainty Principle through modifications of the operators as well as the commutation relations. The show that, depending on the choice of the GUP parameter $\beta$, these modifications either yield the expected minimum length behaviour, or alternatively no minimum length. I find this paper a useful contribution to the existing GUP literature, and I believe it will be of interest to those in the quantum gravity community.

Author Response

We thank referee #1 for their comments. Since there were no critical comments we did not make any changes in connection with this report.

Reviewer 2 Report

Modified dispersion relations between momenta and position operators (e.g. the ones with a minimal length) have been studied in theoretical physics in connection with the so-called quantum gravity effect. They are based on a modification of quantum mechanical canonical commutation relations by changing some operator realization.  For example, the simplest one in momentum realization relies on adding a cubic term to the standard momentum operator leaving the position operator unchanged.

Since there is no experimental/observational evidence for such modification authors propose some alternatives. Their idea is to modify at the same time the realisation of position and momenta living the canonical relations unchanged. This can be easily done by e.g., performing a unitary similarity transformation on the standard realization.  For some examples produced in the paper this can also reproduce a minimal length effect. An extended discussion of the result is provided.

Author Response

We thank referee #2 for their comments. Since there were no critical comments we did not make any changes in connection with this report.

Reviewer 3 Report

09 Feb. 2022 - universe-1605920-report

A Subtle Aspect of Minimal Lengths in the Generalized Uncertainty Principle 
by Michael Bishop, Joey Contreras, and Douglas Singleton

The paper considers a few possible ways of introducing the Generalized Uncertainty Principle (GUP) and whether there would be a minimal length involved. The focus is specifically on the modifications of the commutation relations between the momentum and position operators. A few clear examples are given where the GUP results in a minimal length at a specific momentum spread, no minimal length, or minimal length but within a pair of operators with the standard commutation relations. As such, the paper points at an overlooked and subtle feature of various GUP extensions and the realization of the pairs of modified operators.

The introduction section provides a good overview of the problem, the motivation behind it, and key ideas and results in the literature. The second section is devoted to the example of minimal length but within a pair of operators with the standard commutation relations. Finally, the last - third section of the paper contains the summary and conclusions.

I find the paper to be original and intellectually stimulating and recommend it for publication upon a few additions or modifications to address some of the following concerns:

1) It seems to me that the authors are pre-determined that there should be a minimal length in nature. For example, in lines 6 & 7 of the abstract "... it rules out certain GUPs based on the nonexistence of a minimal length scale." 
It is the experiment that is yet to show that there is a universal minimal length in nature. We are used to seeing characteristic scales in various processes, but these are mostly due to the presence of mass and charge or interplay of kinetic and potential energy. There are well-known examples of conformal invariance as relevant to nature. I suggest stepping back from the "obvious" expectation for minimal length in nature and modifying the paper to be less biased towards an absolute minimum length and more into presenting the problem as a possibility to explore nature from an unbiased perspective.

2) Another bias seems the expectation that the GUP is relevant due to quantum gravity effects. The paper briefly discusses a hand waving argument for the momentum dependence of \Delta{x} (line 20), that helps relate G to \beta in (5). While this seems to be a sensible thing to do, I would like to point out that G is treated as a constant. However, this is not consistent with the expected quantum effects where interaction parameters are subject to renormalization flow depending on the energy scale. I suggest setting the discussion as general as possible and to dedicate a separate section on the estimate of \beta from various viewpoints, interpretations, and potential experimental measurements. After all, nature seems to pull a new trick for us every time we press hard on it with an increase in our experimental energy scale.

3) If the unbiased approach (see 1 & 2 above) is taken, then there is a natural question to be addressed:  When and why it is not possible to have standard commutation relations? What does it prevent one to do so? 
I am concerned that in any laboratory experiment we will eventually have to reevaluate back into the standard commutation relation since it is the definition of canonical coordinates. Note that \hbar becomes a fundamental conversion constant based 
on the canonical form of the operators x & p. Thus, the discussion in section 2 is very relevant but in this case, it is about the quantum state of a system which by (1) get us nowhere. Now the question can be posed as to why only the test Gaussian wave-function in momentum space is to be used here?
What about the eigenfunctions of the corresponding X and P operators in section 2, what would they imply for 
the relationship between \delta{X} and \delta{P}?

4) As seen in 3 above, now the questions are more about the structure of the Hilbert space and the admissible quantum states for a system that may exhibit a GUP behavior. Are there examples of known such systems already?
How is the result in Fig. 3. interpreted in the view of such examples?
What is the physical significance of the behavior shown in Fig. 3 in general?

Minor comments and suggestions: 

1) Could there be a better section title for section 2 (line 74)?

2) line 38: add ", it " in "In contrast to (5) is not clear" -> In contrast to (5), it is not clear... 

3) In the last paragraph of section 3. Summary and conclusions the authors are bringing in non-commutative geometry while it was never before discussed in the paper. It is generally better to have all the references and concepts introduced first in the introduction and then to use the main body of the paper to show and present how these concepts come together and to present arguments in support of the main punch line of the paper. In this respect, the last section is to put all this together in a strong and by then obvious conclusions. So, I suggest making relevant changes to the introduction and the conclusions to align the paper with the usual exposition format.

Author Response

We thank referee #3 for their comments and we list below the changes we have made in response to the criticisms and suggestions. We have copied the referee's comments and made our answers after each comment. 

1) It seems to me that the authors are pre-determined that there should be a minimal length in nature. For example, in lines 6 & 7 of the abstract "... it rules out certain GUPs based on the nonexistence of a minimal length scale." 
It is the experiment that is yet to show that there is a universal minimal length in nature. We are used to seeing characteristic scales in various processes, but these are mostly due to the presence of mass and charge or interplay of kinetic and potential energy. There are well-known examples of conformal invariance as relevant to nature. I suggest stepping back from the "obvious" expectation for minimal length in nature and modifying the paper to be less biased towards an absolute minimum length and more into presenting the problem as a possibility to explore nature from an unbiased perspective.

We agree that quantum gravity may not have a minimal distance; that this is something to be decided by experiment. To this end we have softened that statement in the abstract and throughout the paper we have tried to emphasize that it is experiment that will in the end show what the nature of quantum gravity is. In addition to modifying the statement in the abstract we have also given some more details about the types and nature of experiments that one could do to test quantum gravity/GUP theories. This added material is line 28 through line 80. Some of this material was there before, but we have greatly expanded the details of the types of experiments that could tease out details about QG/GUP. We have also added a reference to a 2012 work by J. Bekenstein on this topic (newly added reference [15]).

2) Another bias seems the expectation that the GUP is relevant due to quantum gravity effects. The paper briefly discusses a hand waving argument for the momentum dependence of \Delta{x} (line 20), that helps relate G to \beta in (5). While this seems to be a sensible thing to do, I would like to point out that G is treated as a constant. However, this is not consistent with the expected quantum effects where interaction parameters are subject to renormalization flow depending on the energy scale. I suggest setting the discussion as general as possible and to dedicate a separate section on the estimate of \beta from various viewpoints, interpretations, and potential experimental measurements. After all, nature seems to pull a new trick for us every time we press hard on it with an increase in our experimental energy scale.

We did not add a new section in regard to this comment but we did put extensive discussion of this point into the conclusion from line 161 through line 179. We point out, as the referee suggested that in QFT coupling are not constant but rather run with momentum-energy scale. We recall the example of the running of the QED coupling via the fine structure “constant”. We point out that line 168 through line 178 that one can interpret the modification of the commutator as a scale dependent β and therefore a scale dependent G. However, we also point out the differences in line 174 through 179 – the running seems quadratic rather than logarithmic and there is work (newly added reference [32] by Anber and Donoghue) that shows it can be tricky to consistently define a running G.  

3) If the unbiased approach (see 1 & 2 above) is taken, then there is a natural question to be addressed:  When and why it is not possible to have standard commutation relations? What does it prevent one to do so?  I am concerned that in any laboratory experiment we will eventually have to reevaluate back into the standard commutation relation since it is the definition of canonical coordinates. Note that \hbar becomes a fundamental conversion constant based on the canonical form of the operators x & p. Thus, the discussion in section 2 is very relevant but in this case, it is about the quantum state of a system which by (1) get us nowhere. Now the question can be posed as to why only the test Gaussian wave-function in momentum space is to be used here?
What about the eigenfunctions of the corresponding X and P operators in section 2, what would they imply for  the relationship between \delta{X} and \delta{P}?

We have clarified why we used in particular form of test wave function (a Gaussian in momentum space). This is simply due to the more complicated behavior between P and p for the GUP arising from the operators in equation (6) versus the GUP arising from the operators in equation (3) which does not modify the momentum operator. This discussion is in line 98 through line 117. Some of this material was in the previous version but we have added/corrected/expanded to address this comment.  

4) As seen in 3 above, now the questions are more about the structure of the Hilbert space and the admissible quantum states for a system that may exhibit a GUP behavior. Are there examples of known such systems already? How is the result in Fig. 3. interpreted in the view of such examples?
What is the physical significance of the behavior shown in Fig. 3 in general?

This is one feature of the GUP models given by equations 9 through 12 and shown graphically by figure 3 that is not so strong as compared to the GUP in figure 1 and from equations 2 through 5 – there is no simply physical argument that lead to the GUP of figure 3. The GUP of figure 1 does have a heuristic argument, which we give in the introduction, for this form for a minimal length. We mention this weakness in line 138 through line 145.

Minor comments and suggestions: 

In regard to these comment we have made the changes.

  • Could there be a better section title for section 2 (line 74)?

We have changed the section title for section 2.

  • line 38: add ", it " in "In contrast to (5) is not clear" -> In contrast to (5), it is not clear... 

We have made this change.

  • In the last paragraph of section 3. Summary and conclusions the authors are bringing in non-commutative geometry while it was never before discussed in the paper. It is generally better to have all the references and concepts introduced first in the introduction and then to use the main body of the paper to show and present how these concepts come together and to present arguments in support of the main punch line of the paper. In this respect, the last section is to put all this together in a strong and by then obvious conclusions. So, I suggest making relevant changes to the introduction and the conclusions to align the paper with the usual exposition format.

We have included a discussion of non-commutative geometry approach to a minimal length in the opening section line 28 through line 45, and we return this this point in the concluding section line 161 through line 162.

Round 2

Reviewer 3 Report

09 March 2022 - universe-1605920-report

It seems that the authors have made their best effort to address my concerns and comments. However, I feel that they have ignored two important suggestions of mine. First, adding a discussion section on the estimate of \beta from various viewpoints,  and second a better introduction and discussion of the non-commutative ideas. 
I still feel that such a stand-alone discussion section is better than the current Summary and conclusions section that contains an important argument on the idea of non-commutative spaces as well as last-minute dropping of new references [32-34] not used anywhere in the discussion. I consider these to be minor changes and corrections since these are a matter of style, but have numbered them below:

A) moving relevant material into a new discussion section on running G (betta) and the non-commutative geometry expectations;

B) Rewriting the Summary and conclusions section to reflect better the main points of the paper and its structure;

C) few formulas need corrections and clarifications: 
Eq. (4): if one is to use (2) in (1) then one will obtain also a term ^2, thus it has to be stated that 
one is considering the center of mass coordinate systems where =0.
Eq. (5) is missing hbar;
Eq. (16) - LHS has two 'i' instead of i & j;
Compute [X_i,X_j] using (13), can't you? It should be just a function of p_i, not an operator, isn't it?

D) Note that Eq.(8) re-expressed in delta{p} gives a term (hbar/betta)/delta{p}^3 and the standard hbar/delta{p};
thus, this affects only the trends of delta{x} as delta{p}->0, so the use of the Gaussian wave-function (line 101) 
could be viewed more as a prelude/toolkit-check for next example in section 2. 

E)About the non-commutativity and the argument that is used to justify GUP near line 21:
E1) Isn't the 'p_M maximum cap' natural given that any realistic experiment has access only to a finite energy source - the energy in a causally connected universe? If one is trying to squeeze this energy into as small a region of space as possible and to stay connected would lead to a black hole, isn't it?
E2) The relation of the Schwarzschild radius to the mass of a black hole does not translate into delta{x} and delta{p} relation but more into delta{x} and delta{E} and from there a non-commutativity 
between time and space since delta{x} and delta{t} are related.
E3) the above may be more related to the string-theory than the standard non-commutativity geometry;
nevertheless, going with expressions (13-15) one can compute [X_i,X_j], can't you?

F) fixing spelling/typos and clarifying:
line 6: add it to "models since distinguishes those" -> "models since it distinguishes those";
line 44: change has to have in "GUP models ... also has this...";
line 66: use tabletop and small-scale instead of "... table top, small scale laboratory ...";
line 79: use Planck-scale physics instead and check other places as well...
line 162: and near Eq. (16), it is not clear if (PX-PX) is an argument of betta or betta is multiplied by a term with P(X) or it is a composition of P and X, can you clarify?
line 168&167: the sentence is not clear as to which commutator is to be considered. 

Author Response

We have put our answers to the referee #3 comments from the 2nd round of reviews below in italics after their comments, which we have repeated for clarity.

It seems that the authors have made their best effort to address my concerns and comments. However, I feel that they have ignored two important suggestions of mine. First, adding a discussion section on the estimate of \beta from various viewpoints,  and second a better introduction and discussion of the non-commutative ideas. 
I still feel that such a stand-alone discussion section is better than the current Summary and conclusions section that contains an important argument on the idea of non-commutative spaces as well as last-minute dropping of new references [32-34] not used anywhere in the discussion. I consider these to be minor changes and corrections since these are a matter of style, but have numbered them below:

A. moving relevant material into a new discussion section on running G (betta) and the non-commutative geometry expectations;

We have moved the discuss of non-commutative geometry and running of G into a separate section – see new added section 3 on page 6 and 7.

B. Rewriting the Summary and conclusions section to reflect better the main points of the paper and its structure;

We have re-written the introduction and conclusion to take this new section into account.

C. few formulas need corrections and clarifications:  (4): if one is to use (2) in (1) then one will obtain also a term ^2, thus it has to be stated that  one is considering the center of mass coordinate systems where =0.
Eq. (5) is missing hbar;
Eq. (16) - LHS has two 'i' instead of i & j; Compute [X_i,X_j] using (13), can't you? It should be just a function of p_i, not an operator, isn't it?

After eqn. 4 we have indicated that this result in for the cae =0 which is generally used in work such as our reference [7] KMM. We have put in the missing factor of hbar in eqn.5 as well as in eqn. 8 where we also missed a factor of hbar. Due to changing of the ordering of our discussion of non-commutative geometry we have moved eqn. (16) earlier so that it is now eqn. (14), we have corrected the index typos (now they are “I’ and “j”) and we have given some more details about the derivation of this coordinate commutator which agrees with the same result given in reference [7].

D. Note that Eq.(8) re-expressed in delta{p} gives a term (hbar/betta)/delta{p}^3 and the standard hbar/delta{p}; thus, this affects only the trends of delta{x} as delta{p}->0, so the use of the Gaussian wave-function (line 101)  could be viewed more as a prelude/toolkit-check for next example in section 2. 

We have included some comments about point D in the discussion after eqn. 8 and in particular in the additions/ alterations in the paragraph right below figure 2.

E. About the non-commutativity and the argument that is used to justify GUP near line 21:
E1) Isn't the 'p_M maximum cap' natural given that any realistic experiment has access only to a finite energy source - the energy in a causally connected universe? If one is trying to squeeze this energy into as small a region of space as possible and to stay connected would lead to a black hole, isn't it?
E2) The relation of the Schwarzschild radius to the mass of a black hole does not translate into delta{x} and delta{p} relation but more into delta{x} and delta{E} and from there a non-commutativity 
between time and space since delta{x} and delta{t} are related.
E3) the above may be more related to the string-theory than the standard non-commutativity geometry; nevertheless, going with expressions (13-15) one can compute [X_i,X_j], can't you?

The cap in momentum and energy applies to the momentum carried by a single particle/field via the relationship E2-p2=m2 (again with c=1). This is the context in which a maximum momentum is meant, which conforms to the usage of works like reference [11]. Also, in this context there are modifications not just to momentum and position operators, but to energy and time as well. Some of the comments added to address the above point in E are found at the bottom of the first paragraph in the introduction and after eqn. 9.  

F. fixing spelling/typos and clarifying: line 6: add it to "models since distinguishes those" -> "models since it distinguishes those";
line 44: change has to have in "GUP models ... also has this...";
line 66: use tabletop and small-scale instead of "... table top, small scale laboratory ...";
line 79: use Planck-scale physics instead and check other places as well...
line 162: and near Eq. (16), it is not clear if (PX-PX) is an argument of betta or betta is multiplied by a term with P(X) or it is a composition of P and X, can you clarify?
line 168&167: the sentence is not clear as to which commutator is to be considered. 

We have fixed or addressed these typos in the revisions. The line number are different since we moved some material from the introduction and conclusion to the new section 3.